# Hand Dexterity Is Associated with the Ability to Resolve Perceptual and Cognitive Interference in Older Adults: Pilot Study

**DOI:** 10.3390/geriatrics8020031

**Published:** 2023-02-27

**Authors:** Marie Schwalbe, Skye Satz, Rachel Miceli, Hang Hu, Anna Manelis

**Affiliations:** 1School of Medicine, University of Pittsburgh, Pittsburgh, PA 15261, USA; 2Department of Psychiatry, University of Pittsburgh, Pittsburgh, PA 15213, USA

**Keywords:** interference resolution, cognitive control, modified Simon task, hand dexterity, aging, falls

## Abstract

The relationship between hand dexterity and inhibitory control across the lifespan is underexplored. In this pilot study, we examined inhibitory control using a modified Simon task. During the task, participants were presented with right- and left-pointing arrows located either on the right or the left parts of the screen. In the congruent trials, the arrow location and direction matched. In the incongruent trials, they mismatched, thus creating cognitive interference. In 50% of trials, the arrow presentation was accompanied by a task-irrelevant but environmentally meaningful sound that created perceptual interference. Hand dexterity was measured with the 9-hole peg test. Significantly faster reaction time (RT) on the modified Simon task (*p* < 0.001) was observed in younger adults, trials with concurrent sound stimuli, and congruent trials. Older adults who reported recent falls had greater difficulty resolving cognitive interference than older adults without recent falls. Hand dexterity significantly moderated the effect of sound on RT, but only in the group of older individuals. Interestingly, older individuals with reduced hand dexterity benefited from concurrent sounds more than those with better hand dexterity. Our findings suggest that task-irrelevant but environmentally meaningful sounds may increase alertness and enhance stimulus perception and recognition, thus improving motor performance in older individuals.

## 1. Introduction

Many daily activities, including driving, shopping, and even social interaction, rely on people’s ability to ignore distracting and misleading stimuli to respond effectively to a situation. Inhibitory control is an executive function that helps allocate attention to goal-relevant stimuli while inhibiting extraneous (goal-irrelevant) information coming in the form of perceptual or cognitive interference. Both perceptual interference, such as irrelevant noises or images [1], and cognitive interference, such as misleading information [2], can distract people’s attention from their intended task and worsen task performance. The ability to direct attention to the relevant information and ignore this interference is termed interference resolution, and it is necessary for inhibitory control [3]. Normal aging is associated with worsening inhibitory control, which is demonstrated by the declining performance with age on tasks requiring interference resolution [4]. In addition to playing an important role for cognitive task performance, inhibitory control is critical for effective motor function, such as hand dexterity and strength, where it is necessary to ignore extraneous information [5]. A deeper understanding of the relationship between interference resolution and motor task performance could guide future diagnostics and therapeutic interventions for neurological and movement disorders.

Precision of hand movements is determined by hand dexterity. Reduced hand dexterity is often observed in older individuals and is associated with difficulty performing activities of daily living, decreased quality of life, and even depression [6]. Hand dexterity can be compromised in certain neurological conditions, such as multiple sclerosis [6], stroke [7,8], diabetic peripheral neuropathy [9], Parkinson’s disease [10], and Alzheimer’s disease [11]. A small number of studies have shown that hand dexterity is directly related to cognitive function [12,13,14]. For example, better executive function and attentional control were associated with better hand dexterity in healthy community-dwelling older adults [13,14]. Furthermore, better attentional control was associated with better motor dexterity among stroke survivors and patients discharged from the ICU [15].

While these findings are consistent with the idea that decreased motor function in older individuals is associated with underlying reduced cognitive function, the age-related changes in cognitive and perceptual interference resolution, as well as the relationship between interference resolution and hand dexterity, remain underexplored. The current pilot study aims to fill these gaps and explore interference resolution in older and younger individuals using the modified Simon task [16].

During the modified Simon task, participants were presented with right- and left-pointing arrows either in the right or left part of the screen (cognitive interference). A total of 50% of the trials were accompanied by an everyday sound stimulus that was irrelevant to the task but environmentally meaningful, such as birds chirping or thunder (perceptual interference). The traditional version of the Simon task presents participants with congruent and incongruent either visual or auditory stimuli. Our modified Simon task revisits the effects of visual cognitive interference and explores how interference resolution is affected by task-unrelated but environmentally plausible auditory stimuli. This pilot project will allow us to examine the study protocol and the validity of the modified task by replicating the congruency effects observed in previous studies [16,17,18].

We hypothesized that older individuals would be slower and less accurate than younger individuals and that participants would be slower and less accurate on incongruent trials, especially those with concurrent perceptual interference. In addition, we hypothesized that the individuals with reduced hand dexterity would show greater worsening of reaction time (RT) and accuracy on the interference resolution vs. the no interference resolution trials. This effect would be especially pronounced in older, compared to younger, individuals on the trials that required resolution of both cognitive and perceptual interference. Our alternative hypothesis was that concurrent sound presentation would improve task performance independently of hand dexterity. This latter hypothesis was based on the “pip and pop” effect, which suggests that concurrent perceptual stimuli presentation, such as visual or auditory stimuli, may improve task performance [19,20].

## 2. Materials and Methods

### 2.1. Participants

The study was approved by the University of Pittsburgh Institutional Review Board (IRB number STUDY20120048). Written informed consent was obtained from all participants. Eighty-nine participants between 18–85 years of age were recruited from the community, online Pitt + Me and Pepper (IRB number STUDY19090270) registries, as well as from the ongoing NIH-funded study (R01MH114870). Potential participants either called or emailed the research team, expressing their interest in the study, or they were approached by the study team after the person was identified as potentially eligible through the registry. Participants were fluent in English and had premorbid IQ > 85 per the National Adult Reading Test [21]. Exclusion criteria included a history of head injury, neurodevelopmental and neurological disorders, learning disability, and psychiatric disorders other than depressive and anxiety disorders. We excluded from the analyses the individuals who were diagnosed with autism spectrum disorder, bipolar disorder, and schizophrenia (*n* = 3), and those who were missing the hand dexterity data (*n* = 4), were left-handed or ambidextrous (*n* = 7), or did not understand the task instructions (*n* = 2). This left 73 individuals in the analyses. Our sample included only right-handed participants to avoid the potential bias in right hand vs. left hand responses when the right hand is dominant for some individuals, but non-dominant for others [22,23,24].

### 2.2. Study Procedures

Participants completed intake interviews and self-report questionnaires, cognitive and neurological assessments, a hand dexterity test, and a computerized modified Simon task during an in-person office visit. Participants were paid for participation.

#### 2.2.1. Demographics, Cognitive and Neurological Assessments

Intake interviews and self-reports involved the collection of information about general demographics, health, and current medications. A basic neurological examination of cranial nerves, gait, posture, balance, and sensation was administered by a trained team member to screen for possible neurological deficits. After that, the Montreal Cognitive Assessment (MoCA) [25] was used to assess general cognitive functioning across the core domains of cognition. Visual acuity and age-related macular degeneration (AMD) were assessed with the Snellen test and the Amsler grid accordingly. Participants also reported fall history for the past year.

#### 2.2.2. Hand Dexterity Assessment

The gold standard measurement of hand dexterity is the 9-hole peg test (NHPT) [26,27], which can be used across a wide range of ages and medical conditions [28,29,30].

In this task, the plastic peg board was placed directly in front of the participant, with the dish holding the pegs placed closest to the hand being tested. The participant was instructed to use one hand to take the pegs from the dish, one at a time, and place them into each of the 9 holes on the board as quickly as possible. As soon as all holes were filled, participants removed the pegs one at a time and placed them back into the dish. Participants could fill and empty the holes in any order they choose. The hand that was not being evaluated could be used to steady the board. Performance on the NHPTwas measured using a stopwatch to record the amount of time taken to complete the task in seconds. Recordings were started when the participant touched the first peg and were stopped when the last peg entered the dish. The participants were given an opportunity to practice this task once before the timed trial.

### 2.3. Modified Simon Task

The Simon task [16] was modified to measure participants’ ability to resolve cognitive, perceptual, and combined interference (Figure 1). During this task, participants were shown an image of an arrow pointing either to the left or to the right. They were instructed to indicate which direction the arrow was pointing by pressing the “Z” key on a standard QWERTY keyboard with their left index finger for left-pointing arrows and the “M” key with their right index finger for right-pointing arrows as quickly and accurately as possible.

The task consisted of 2 runs of 32 trials, whose duration was randomly determined and varied from 6500–10,000 ms in 500 ms increments. Each trial consisted of the screen with a fixation star, the stimulus screen, and an inter-trial interval (Figure 1). The fixation star duration was randomly determined for each trial and was between 500–2000 ms. A stimulus presentation was equal to the participant’s reaction time (RT), but was not longer than 2500 ms. The trials were separated with inter-trial intervals (it is), during which participants were shown a “Please Rest” screen. The ITI duration varied from trial to trial and was equal to the time necessary to complete the duration of the trial.

The stimuli were white arrows presented on a black background to the right or to the left of a fixation cross. Of 64 trials, 50% were congruent, and 50% were incongruent. Stimulus congruency refers to the relationship between the arrow location and its direction. In the congruent trials, the arrow location and direction matched (left-pointing arrow presented on the left side of the screen), while in the incongruent trials, the arrow location and direction mismatched (left-pointing arrow on the right side of the screen), thus creating a source of cognitive interference.

A concurrent auditory stimulus that could be a natural (birds chirping or a thunderstorm) or a man-made (construction or sirens) sound accompanied 50% of congruent and 50% of incongruent trials. We believe that presenting unrelated auditory stimuli during the task created perceptual interference similar to that which people experience in their everyday lives. The sounds were found on the Internet and modified to have equal loudness across all sound clips. Each sound clip started simultaneously with the onset of the visual stimulus and ended when a participant responded in the trial. Prior to task completion, sounds were tested with each participant to ensure audibility. Volume was adjusted accordingly based on individual needs.

Each run had an equal number of congruent/incongruent and sound/no sound trials. The order of the congruent/incongruent trials with and without sound, as well as the trial and fixation durations, was randomized for each participant to avoid systematic bias.

### 2.4. Data Analyses

All statistical analyses were conducted using R (https://www.r-project.org (accessed on 24 January 2023)).

#### 2.4.1. Demographic and Hand Dexterity Data Analyses

We calculated the means and standard deviations for the demographic and hand dexterity data across all participants. Considering the wide age range in the study, we median-split the sample (median age = 58.9 years) into two groups. The individuals whose age was below the median comprised a group of younger adults (*n* = 36), while those who were at or above the median age comprised the group of older adults (*n* = 37). Demographic, clinical, and hand dexterity variables were then compared between the younger and older adult groups using chi-square and *t*-tests.

#### 2.4.2. Modified Simon Task Data Analyses

To understand the effects of age and hand dexterity, in addition to the effects of congruency and sound, on RT and accuracy, we utilized mixed-effects models using the lme4 package in R [31]. Two models were examined: a congruency-by-sound-by-age group interaction model and a congruency-by-sound-by-age group-by hand dexterity interaction model. A significance level *p* < 0.05 was utilized for all statistical analyses, except for the analyses that included hand dexterity. Those analyses were conducted for both hands, so the significance level was adjusted to *p* < 0.025 to reflect Bonferroni correction (0.05/2 = 0.025). In all models, participants were treated as a random factor, and participants’ IQ was used as a covariate. The contrasts and means were estimated from the mixed-effects models using the ‘modebased’ package in R [32]. The *p*-values, ANOVA, and summary tables were produced using the lmerTest package [33].

RT analysis used only trials with accurate responses. Before entering RT values in the analysis, they were examined for outliers. The values that were outside the 3 IQRs (interquartile ranges) from the first or third quartile were considered as outliers and were excluded from the analyses. RT was fitted using linear mixed-effects models (the *lmer* function).

Accuracy was analyzed as a binomial variable (1 for correct responses, 0 for incorrect responses). It was analyzed using generalized linear mixed-effects models (the *glmer* function for binomial data), and a Wald’s chi-square and *p*-values were reported.

## 3. Results

### 3.1. Demographic and Clinical

The demographics and clinical characteristics of the participants are reported in Table 1. The groups of younger and older participants did not differ in terms of sex, general cognitive function per MoCA, or neurological health. As expected, older adults were significantly older than younger adults (*p* < 0.001). Older adults had higher IQ (*p* < 0.05), but slower performance on the hand dexterity task in both the dominant and non-dominant hand (*p* < 0.01) compared to younger adults. Across all participants, the dominant hand dexterity responses were faster than those in the non-dominant hand (t(72) = −6.9, *p* < 0.001). In the group of older, but not younger, participants, 16 individuals (43%) were diagnosed with AMD, and 7 (19%) reported a history of falls during the past 12 months.

### 3.2. Behavioral

#### 3.2.1. The Effect of Task Condition and Age Group on RT and Accuracy in the Modified Simon Task

Based on the IQR analysis of RT, 1.2% of responses were outliers and were excluded from the analyses.

The mixed-effects analysis of the congruency-by-sound-by-age group interaction effect revealed a significant congruency-by-age group interaction [F(1, 4370.1) = 12.7, *p* < 0.05], as well as the main effects of congruency [F(1, 4376.6) = 273.9, *p* < 0.001], sound [F(1, 4376.1) = 24.9, *p* < 0.001] and age group [F(1, 70) = 33.5, *p* < 0.001] on RT (Figure 2). Older adults were slower than younger ones (z = 5.8, *p* < 0.001). Participants were slower on trials without sound than those with sound (z = 4.99, *p* < 0.001), and they were slower on incongruent trials than congruent trials (z = 16.55, *p* < 0.001).

The analysis of accuracy revealed significant main effects of age group (chi2 = 4.3, *p* < 0.05) and congruency (chi2 = 28.9, *p* < 0.001). Older participants had lower accuracy than younger participants (z = −2.0, *p* < 0.05). The accuracy for congruent trials was higher than that for the incongruent trials (z = 5.4, *p* < 0.001). Table 2 reports the mean RT and accuracy values that were estimated from the mixed-effects models.

#### 3.2.2. The Effect of Task Condition, Hand Dexterity, and Age Group on RT in the Modified Simon Task

The mixed-effects analysis for the dominant (right) hand dexterity revealed a significant sound-by-right hand and dexterity-by-age group interaction effect [F(1, 4370.1) = 12.7, *p* < 0.001] that survived the Bonferroni correction for 2 tests. In addition, there were significant interaction effects between sound and right hand dexterity [F(1, 4370.1) = 4.5, *p* = 0.033], sound and age group [F(1, 4370.1) = 14.4, *p* < 0.001], and sound and congruency [F(1, 4370.2) = 4.4, *p* = 0.036], and a main effect of congruency [F(1, 4370.6) = 4.9, *p* = 0.027]. Of these interaction effects, only sound-by age group interaction survived Bonferroni correction. Further exploration of these effects in each age group revealed a main effect of congruency [F(1, 2197.1) = 6.9, *p* < 0.01], with faster responses for congruent vs. incongruent trials [model estimated t(2197.09) = −11.74, *p* < 0.001], but no effects of sound or hand dexterity in the younger individuals. In contrast, in older adults, there was a significant sound-by-right hand dexterity interaction effect [F(1, 2173.04) = 15.5, *p* < 0.001], suggesting that individuals with reduced hand dexterity benefited from concurrent sound presentation more than those with faster hand dexterity (Figure 3). There was also a main effect of sound [F(1, 2173.05) = 12.0, *p* < 0.001], with faster responses on the trials with sound compared to the trials without sound [model estimated t(2173.12) = 2.71, *p* < 0.01].

The mixed-effects analysis for the non-dominant (left) hand dexterity revealed a significant sound and age group interaction [F(1, 4370.1) = 4.26, *p* = 0.04] that, however, did not survive Bonferroni correction. Further exploration of these effects in each age group revealed no significant interactions or main effects in younger individuals. In older adults, there was a significant sound-by-left hand dexterity interaction effect [F(1, 2173.03) = 6.9, *p* < 0.01], suggesting that individuals with reduced hand dexterity benefited from concurrent sound presentation more than those with faster hand dexterity (Figure 3). There was also a main effect of sound [F(1, 2173.05) = 4.5, *p* = 0.033], with faster responses on the trials with sound compared to the trials without sound [model estimated t(2173.12) = 2.72, *p* < 0.01].

#### 3.2.3. The Effect of Task Condition, Hand Dexterity, and Age Group on Accuracy in the Modified Simon Task

Performance accuracy was above 95% in all conditions and groups (Table 2). The mixed-effects analysis for the dominant (right) hand dexterity revealed a significant congruency-by-right hand dexterity interaction effect [chi2(1) = 4.1, *p* = 0.043] on the accuracy, with greater differences between accuracy for congruent and incongruent trials observed in those with reduced dominant hand dexterity. This effect did not survive Bonferroni correction, however.

The mixed-effects analysis for the non-dominant (left) hand dexterity revealed no significant main or interaction effects of left hand dexterity with the other variables.

#### 3.2.4. Exploratory Analyses in the Group of Older Individuals

We examined the effects of AMD status (diagnosed vs. not diagnosed with AMD) and recent falls (reported falling vs. not falling during the past 12 months) on RT in the group of older adults. The mixed-effects model that examined the sound-by-congruency-by-AMD status interaction revealed no significant main effects of AMD or any interaction with AMD status on RT in the modified Simon task.

The mixed-effects model that examined the sound-by-congruency-by-recent falls status interaction revealed a significant congruency-by-falls interaction effect [F(1, 4370) = 4.26, *p* < 0.05]. The difference between the congruent and incongruent conditions was greater for those older adults who reported recent falls vs. those who did not report falling during the past 12 months (no falls: incongruent-congruent difference = 65 msec (SE = 7.5 msec); falls: incongruent-congruent difference = 150.2 (SE = 15.7 msec); Figure 4).

## 4. Discussion

In this study, we used a modified Simon task to investigate the effects of cognitive and perceptual interference on the task’s RT and accuracy in neurologically and cognitively normal older versus younger adults. Consistent with previous studies [16,34,35,36], participants’ responses were slower and less accurate on the incongruent, compared to congruent, trials, thus supporting the validity of the modified Simon task. Older adults’ responses were slower and less accurate compared to those of younger adults, reflecting an age-related decline in task performance [4]. Considering that the accuracy in the modified Simon test was very high (almost at ceiling), we will not further discuss the accuracy findings due to their low clinical relevance.

Inconsistent with our predictions that perceptual interference would worsen RT, we found that irrelevant but environmentally meaningful sounds presented concurrently with visual stimuli improved RT in both congruent and incongruent trials across all participants. These findings are consistent with the “pip and pop” effect, suggesting that nonspatial, task-irrelevant auditory stimuli may improve performance on visual tasks, including those requiring visual search [19,20]. Although previous studies suggested that the effect of meaningless background noise on cognitive performance depends on the intensity, duration, timing, and type of noise, along with the subjects’ age and the task load [37,38,39], we found benefits of sound in both older and younger adult age groups, as well as in both congruent (easier) and incongruent (more difficult) trials. Our findings may be explained by the nature of the sounds in our study. Even though the sounds were irrelevant to the task at hand, they were environmentally meaningful in that they carried important information about the surrounding environment. For example, if a person hears a screeching sound of a car brake, they should quickly orient their attention toward this sound (even if they are engaged in some other activity) to evaluate the level of danger and produce a quick response if needed.

The other goal of this study was to understand how hand dexterity is related to participants’ ability to resolve interference. Consistent with previous studies, reduced hand dexterity was observed in older, compared to younger, individuals [13,40] and for the non-dominant, compared to the dominant, hand [41]. Despite the latter differences, we found that hand dexterity in both dominant and non-dominant hands moderated the effect of sound on RT in the modified Simon task, but only in the group of older individuals. Interestingly, older individuals with reduced hand dexterity benefited from the sound administration more than those with faster dexterity, suggesting a possible implication of these results to improve function in people with movement disorders, such as Parkinson’s disease or multiple sclerosis. Reduced hand dexterity in older adults may be explained by the age-related reduction in strength and muscle mass [42], changes in connective tissues, decrease in number of motor neurons, conditions such as osteoarthritis or rheumatoid arthritis [43,44,45], and a decline in attentional control [5]. Administering the task-irrelevant but environmentally meaningful sounds concurrently with the visual stimuli may increase alertness in older individuals and thus improve motor performance. Given the link between attentional control and motor control [5], it is possible that changes in attention, such as those induced by sound stimuli, are more prominent in individuals with reduced hand dexterity and in individuals with reduced attention (e.g., older adults). Notably, a similar sound effect was observed in children with and without Attention Deficit Hyperactivity Disorder, whose attentional performance was facilitated by administration of task-irrelevant sounds [46]. Lastly, considering that sounds improve visual perception of stimuli [47,48], the sound-related RT improvement in older adults with reduced dexterity could be due to enhanced stimulus perception and faster recognition of whether the arrow was pointing to the right or to the left.

The exploratory analyses conducted in the group of older adults showed that those individuals who reported falls within the past 12 months had greater difficulty resolving cognitive interference (i.e., had a greater difference in RT between congruent and incongruent trials) than those individuals who did not report recent falls. With more than 3 million older adults receiving treatment for fall injuries annually [49], understanding cognitive processes that might be associated with increased risk of fall is critically important. Traditionally, it is thought that falls in older adults are associated with the use of various medications, including psychotropics [50], and worsening of balance and gait patterns [51], as well as visual and cognitive decline [52]. One clinical implication of out pilot study (albeit in the small sample) is the improved understanding that a reduced ability to resolve cognitive interference may be another risk factor for falls, even in neurologically and cognitively normal older adults. These results warrant further prospective examination of the relationship between falls in older adults and their ability to resolve cognitive and perceptual interference (e.g., the person fell because they talked to a friend and had not noticed an obstacle in the hallway).

With regard to visual disturbance, AMD is one of the most prevalent causes of visual impairment and affects approximately 19.8 million Americans aged 40 and older [53]. In our pilot study, we compared a small sample of older adults with AMD vs. those without. We found no significant differences in RT or accuracy between these groups in either task condition. While the small sample size did not allow us to derive a strong conclusion regarding the effects of AMD on task performance, the findings suggest that performance on cognitive tasks is not necessarily affected by an AMD diagnosis if it is not debilitating and if older adults are neurologically and cognitively healthy.

This pilot study’s limitations include the small sample size and cross-sectional approach. Further, all our participants were neurologically healthy, which did not allow us to extrapolate whether the sound administration would be beneficial for older individuals with neurological and movement disorders. Future research should examine the effect of sound on motor response and cognitive/perceptual interference resolution in patients with various neurological disorders (e.g., Alzheimer’s disease, Parkinson’s disease, stroke, and multiple sclerosis), as well as cognitive impairments. For example, approximately 60% of multiple sclerosis patients report impaired hand function in the first year after diagnosis, which makes performance of activities of daily living more difficult [54]. Future studies should examine whether administering task-irrelevant but environmentally meaningful sounds may improve hand function in affected individuals by altering the level of alertness. Neuroimaging studies of the sound effect in older adults may help uncover the neural mechanisms underlying the observed behavior change.

## 5. Conclusions

In conclusion, our findings demonstrated that sound presentation improved performance on the modified Simon task independently of cognitive interference across all participants. The effect of sound on RT was moderated by participants’ hand dexterity. However, this effect was only observed in older individuals, of whom participants with reduced hand dexterity benefited from concurrent sound presentation more than their peers with faster hand dexterity. Taken together, these findings warrant further exploration of the role of environmental sounds on cognitive tasks performance as a therapeutic intervention that can be administered along with other treatments for improving hand function, gait, and fall prevention in older adults [55,56].

## Figures and Tables

**Figure 1 geriatrics-08-00031-f001:**
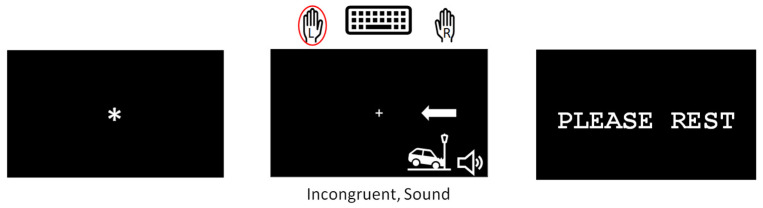
Example of one trial of the modified Simon task.

**Figure 2 geriatrics-08-00031-f002:**
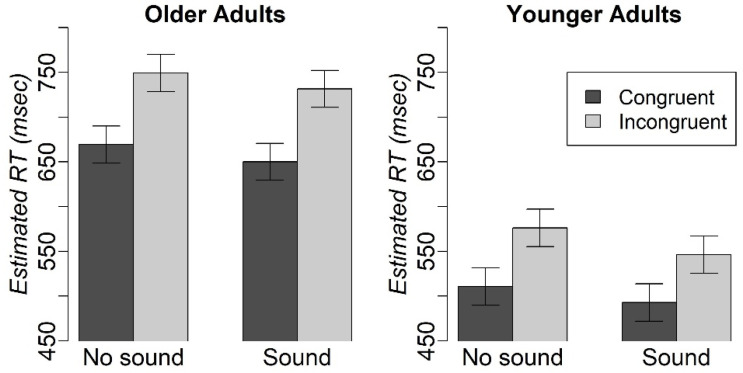
RT on accurate trials in the modified Simon task in older and younger adults. Mean RT and Accuracy in the modified Simon task.

**Figure 3 geriatrics-08-00031-f003:**
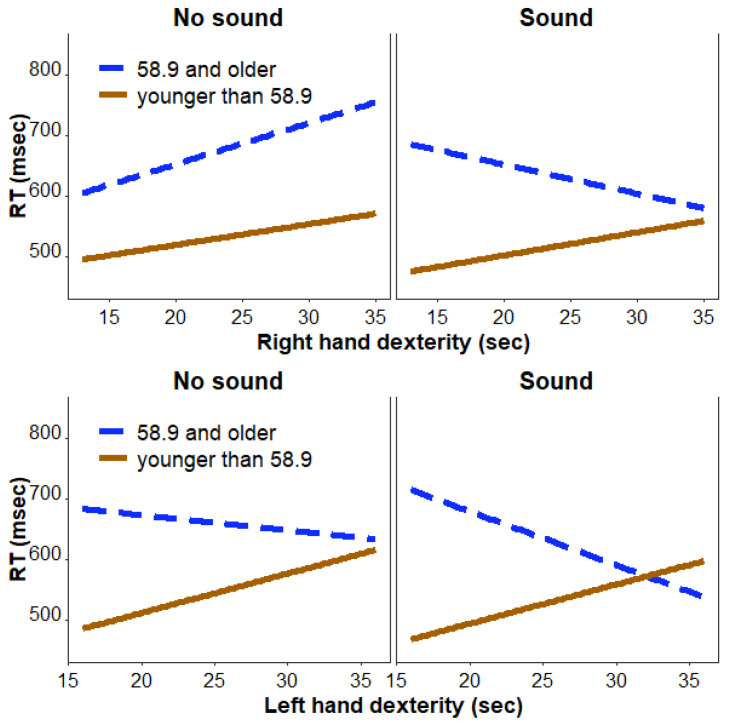
The relationship between hand dexterity and the sound condition in older and younger adults.

**Figure 4 geriatrics-08-00031-f004:**
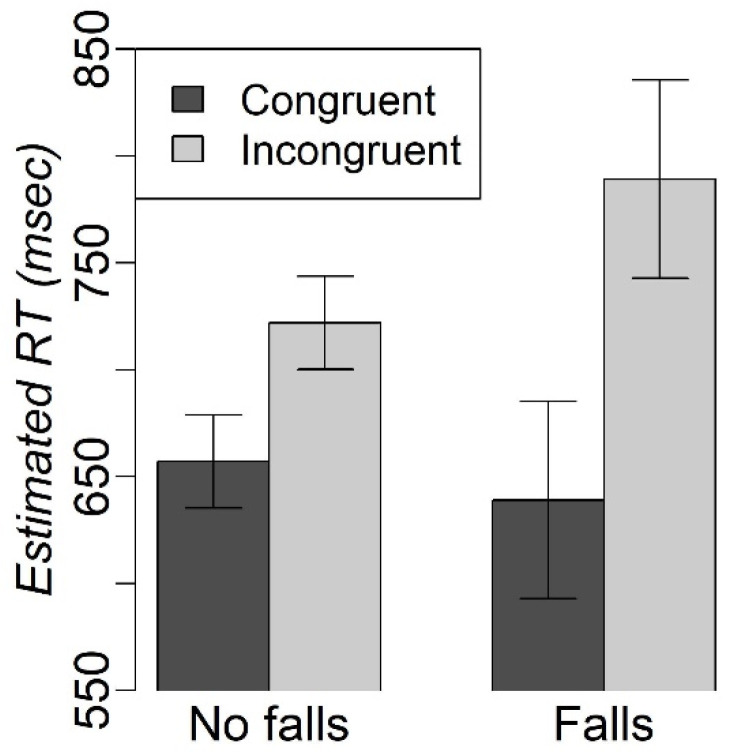
The effect of congruency and recent falls on RT in the modified Simon task.

**Table 1 geriatrics-08-00031-t001:** Demographics information across all participants and between age groups.

	17.2	Older Adults (58.9–85 yo)	Younger Adults (18–58.9 yo)	Statistics for Comparison of Older vs. Younger Adults	Cohen’s d for Comparison of Older vs. Younger Adults
	Mean (SD)	Mean (SD)	Mean (SD)		
*n*	73	37	36		
Sex (number of females)	42	21	21	chi2 = 0, *p* = 1	na
Age (years)	53.33(20.76)	71.61(6.32)	34.54(11.36)	t(71) = 17.29 *p* < 0.001	4.1
IQ (NART)	112.23(6.82)	114.2(5.58)	110.21(7.43)	t(71) = 2.6 *p* = 0.01	0.62
Cognitive Function (MoCA)	27.52(1.82)	27.3(2.09)	27.75(1.48)	t(71) = −1.06 *p* = 0.3	−0.25
Right (Dominant) Hand Dexterity (sec)	20.27(3.45)	21.54(3.52)	18.97(2.88)	t(71) = 3.41 *p* = 0.001	0.81
Left (non-Dominant) Hand Dexterity (sec)	22.33(3.76)	23.95(4.12)	20.67(2.46)	t(71) = 4.12 *p* = 0.0001	0.98
AMD Status	16	16	0	na	na
(Number AMD)					
Fall Status (Number with fall)	7	7	0	na	na

**Table 2 geriatrics-08-00031-t002:** Model estimated RT and accuracy in the modified Simon task.

Congruency	Sound Condition	Age Group	RT, Msec Mean(SD)	Accuracy Mean(SD)
Congruent	No sound	older	657.36(192.94)	0.98(0.14)
Congruent	No sound	younger	513.78(163.27)	0.99(0.11)
Congruent	Sound	older	640.21(200.77)	0.98(0.14)
Congruent	Sound	younger	498.04(160.64)	1(0.06)
Incongruent	No sound	older	737.33(200.07)	0.96(0.19)
Incongruent	No sound	younger	581.16(180.43)	0.96(0.19)
Incongruent	sound	older	721.13(192.17)	0.95(0.22)
Incongruent	sound	younger	550.73(166.11)	0.96(0.2)

## Data Availability

The datasets used and/or analyzed during the current study are available from the corresponding author on reasonable request.

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
