# Peer review of "Hand Dexterity Is Associated with the Ability to Resolve Perceptual and Cognitive Interference in Older Adults: Pilot Study"

_geriatrics, 2023, doi:10.3390/geriatrics8020031_

Round 1

Reviewer 1 Report

This is a well-designed and performed study. I only the following minor comments:

Lines 137-139: The definition of ITI here is a bit confusing, as it's supposed to reflect the cycle at which trials are repeated. The authors should remove it from here, as it also seems unnecessary.

Lines 200-211: It's not clear what statistical tests the authors conducted, and why they report F ratios for one and Chi-squares for the other variable. More info is warranted in the Methods or Results.

Results: Given the authors repeated the same analyses for both hands, they should have made multiple-comparison corrections, or should explain why these corrections were unwarranted.

Author Response

We thank reviewers for their comments. Below we provide detailed point-by-point responses and accompanying revisions to address reviewers’ comments. 

Reviewer 1

Q1.1 Lines 137-139: The definition of ITI here is a bit confusing, as it's supposed to reflect the cycle at which trials are repeated. The authors should remove it from here, as it also seems unnecessary.

A1.1 ‘ITI’ stands for inter-trial intervals. Describing the ITI values is important for better task description, understanding, replicability, and interpretation.  In fact, reporting ITIs is a common practice in psychology research. In the revised manuscript we provide more detailed description of the task at the trial and block level.

“The task consisted of 2 runs of 32 trials whose duration was randomly determined and varied from 6500-10000ms in 500ms increments. Each trial consisted of the screen with a fixation star, the stimulus screen, and inter-trial interval (Figure 1). A fixation star duration was randomly determined for each trial and was between 500-2000ms. A stimulus presentation was equal to participant’s reaction time (RT) but was not longer than 2500ms. The trials were separated with inter-trial intervals (ITI) during which participants were shown a “Please Rest” screen. The ITI duration varied from trial to trial and was equal to the time necessary to complete the duration of the trial.” ~ p. 4 of the revised manuscript.

Q1.2 Lines 200-211: It's not clear what statistical tests the authors conducted, and why they report F ratios for one and Chi-squares for the other variable. More info is warranted in the Methods or Results.

A1.2 Thank you for requesting a clarification for the statistical analyses. RT is a continuous variable, while the accuracy is a binomial variable. The anova() for lmer (for continuous DV) and Anova() for glmer (for monomial DV) produce the results of the omnibus test for fixed effects using Type III SS. The output of the lmer/anova() reports F- and p-values, while the output of the glmer/Anova() reports a Wald’s chi-square and p-values. In the revised manuscript, we clarified this on p 5:

RT analysis. Only RTs on correctly answered trials were used for these analyses. Before entering RT values to the analysis they were examined for outliers. The values that were outside the 3 IQRs (interquartile range) from the first or third quartile were considered as outliers and were excluded from the analyses. RT was fitted using linear mixed effects models (the lmer function).”

Accuracy analyses. The accuracy is a binomial variable (1 for correct responses, 0 for incorrect responses). It was analyzed using generalized linear mixed-effects models (the glmer function for binomial data) and a Wald’s chi-square and p-values were reported”

Q1.3 Results: Given the authors repeated the same analyses for both hands, they should have made multiple-comparison corrections, or should explain why these corrections were unwarranted.

A1.3 We agree with the reviewer. The revised manuscript now includes a line suggesting that “A significance level p<0.05 was utilized for all statistical analyses except for the analyses that included hand dexterity. Those analyses were conducted for both hands, so the significance level was adjusted p<0.025 to reflect Bonferroni correction (0.05/2=0.025).” Throughout the results section we indicated whether or not a particular analysis that used hand dexterity as an independent variable survived or not survived Bonferroni correction for two tests. We also found errors in the section 3.2.3, which were addressed in the revised manuscript. 

Reviewer 2 Report

Thank you for allowing me to review this interesting manuscript. I have the following few comments.

2. Why did you state that it is a pilot study? Can you explain that? 

1. How was the sample calculated? And was it enough for the statistical analysis used? 

2. provide more details on the study procedure. e.g., how did you approach the target population and recruit the sample? etc.

3.  Where are the reliability and validity tests for the modified Simon task? And why did you modify it? 

4. Why is your neurological assessment limited to MoCA, visual acuity, and macular degeneration?  

5. I am unfamiliar with the software you used to analyze the data. Can you provide more details? 

6. Where are the study implications? 

Author Response

We thank reviewers for their comments. Below we provide detailed point-by-point responses and accompanying revisions to address reviewers’ comments. 

Reviewer 2

Q2.1 Why did you state that it is a pilot study? Can you explain that? How was the sample calculated? And was it enough for the statistical analysis used? 

A2.1 We call this study a pilot study because we used it as an opportunity to test the study protocol and to validate the modified Simon task. As there was no pilot data available, we used the general convention that the sample with n>30 is needed for the central limit theorem to hold. Although we had only 73 people in the analyses, the mixed effects models use all observations in all participants for the analyses. So, for example, the model “Sound * Congruency * Right hand dexterity *Age group” used 4455 observations  to predict RT from a total of 15 variables including main effects and interactions (Sound Condition, Sound Condition:Congruency, Sound Condition:Congruency:Right hand dexterity, Sound Condition:Congruency: Right hand dexterity:age group, etc). Considering that a common rule is to have 30 subjects per one regressor and then add 10 more subjects per each additional regressor, it seems like we hadsufficient power for the analyses we conducted.

Q2.2 provide more details on the study procedure. e.g., how did you approach the target population and recruit the sample? etc.

A2.2 We have added a few additional details regarding recruitment approaches and study procedures on p. 2 of the revised manuscript:

“Eighty-nine participants between 18-85 years of age were recruited from the community, online Pitt+Me and Pepper (IRB number STUDY19090270) registries as well as from the ongoing NIH-funded study (R01MH114870). Potential participants either called or emailed to the research team expressing their interest in the study, or they were approached by the study team after the person was identified as potentially eligible through the registry.”

Q2.3 Where are the reliability and validity tests for the modified Simon task? And why did you modify it? 

A2.3. This pilot study did not aim to formally measure validity and reliability of the modified Simon task as we did not believe that adding task-unrelated sounds would significantly affect either of these characteristics. The revised manuscript specifies on p. 2 that

“During the modified Simon task, participants were presented with the right- and left-pointing arrows either in the right or left parts of the screen (cognitive interference). 50% of the trials were accompanied by an everyday sound stimulus that were irrelevant to the task but environmentally meaningful, such as birds chirping or thunder (perceptual interference). The traditional version of the Simon task presents participants with congruent and incongruent either visual or auditory stimuli. Our modified Simon task revisits the effects of visual cognitive interference and explores how interference resolution is affected by task-unrelated but environmentally plausible auditory stimuli. This pilot project will allow us to examine the study protocol and the validity of the modified task by replicating the congruency effects observed in previous studies [16–18].”

The discussion section how indicates that “Consistent with previous studies [16,34–36], participants’ responses were slower and less accurate on the incongruent, compared to congruent, trials thus supporting the validity of the modified task.”

Q2.4. Why is your neurological assessment limited to MoCA, visual acuity, and macular degeneration?  

A2.4 We appreciate your feedback and sorry that our writing was not clear. In addition to MoCA, visual acuity, and macular degeneration, we did perform a standard neurological assessment that examined cranial nerves, gait, posture, balance, and sensation. The goal of this neurological assessment was to detect gross neurological deficits that would exclude the patient from the study. We clarified this information on p. 3 of the revised manuscript:

“Intake interviews and self-reports involved the collection of information about general demographics, health, and current medications. A basic neurological examination that examined cranial nerves, gait, posture, balance, and sensation was administered by a trained team member to screen for possible neurological deficits. After that, the Montreal Cognitive Assessment (MoCA) [25] was used to assess general cognitive functioning across the core domains of cognition.”

Q2.5. I am unfamiliar with the software you used to analyze the data. Can you provide more details? 

A2.5 R is a free software environment for statistical computing and graphics akin to Python, Matlab, and SPSS which is widely used by the scientific community for the past 20 years. For more information about the R software, we would like to respectfully refer the reviewer to the link we reference in the paper on p. 4:

“All statistical analyses were conducted using R (https://www.r-project.org).”

Q2.6 Where are the study implications? 

A2.6 Thank you for this question. This study has potential implications for clinical care and future research. Below we enumerate the study implications as they are reported in the revised manuscript:

Lines 339-348: “Interestingly, older individuals with reduced hand dexterity benefited from the sound administration more than those with faster dexterity suggesting a possible implication of these results to improve function in people with movement disorders, such as Parkinson’s disease or multiple sclerosis. Reduced hand dexterity in older adults may be explained by the age-related reduction in strength and muscle mass [42], changes in connective tissues, decrease in number of motor neurons, conditions such as osteoarthritis or rheumatoid arthritis [43–45], and a decline in attentional control [5]. Administering the task-irrelevant but environmentally meaningful sounds concurrently with the visual stimuli may increase alertness in older individuals and thus improve motor performance.”

Lines 366-372: “One clinical implication of out pilot study (albeit in the small sample) is the improved understanding that a reduced ability to resolve cognitive interference may be another risk factor for falls even in neurologically and cognitively normal older adults. These results warrant further prospective examination of the relationship between falls in older adults and their ability to resolve cognitive and perceptual interference (e.g., the person fell because they talked to a friend and had not noticed an obstacle in the hallway).”

Lines 389-393: “Future studies should examine whether administering task-irrelevant but environmentally meaningful sounds may improve hand function in affected individuals by altering the level of alertness.  Neuroimaging studies of the sound effect in older adults may help uncover the neural mechanisms underlying the observed behavior change.”

Round 2

Reviewer 2 Report

The authors have addressed all my comments adequately. 

Author Response

Thank you very much for your helpful comments